**Data Availability Statement:** All relevant data and syntaxes are within the paper and its Supporting Information files.

# The patient satisfaction in primary care consultation—Questionnaire (PiC): An instrument to assess the impact of patient-centred communication on patient satisfaction

**Stefanie Stark**[ID]*, **Lukas Worm, Marie Kluge, Marco Roos**[ID]**, Larissa Burggraf**

Institute of General Practice, Friedrich-Alexander-University Erlangen-Nürnberg (FAU), Erlangen, Germany

* Stefanie.Stark@uk-erlangen.de

## Abstract

### Background

Primary care consultation is significantly influenced by communication between the General Practitioner (GP) and their patients. Hypothesising that patient satisfaction can be tested based on an expectation-experience comparison, the aim of this article is to discuss the influence of communication on patient satisfaction.

### Methods

A standardised questionnaire was developed striving for a universal primary care survey tool that focuses on patient satisfaction in the context of patient-centred-communication. The sample consisted of 14 German GPs with 80 patients each (n = 1120). Due to the inclusion in an overarching cluster-randomised-study (CRT), the medical practices to be examined were divided into intervention and control groups. The intervention was developed as a reflective training on patient-centred communication.

### Results

The results in the present sample show no correlation between patient-centred-communication and patient satisfaction. There are also no significant differences between the intervention and control group.

### Discussion

The results raise the question to what extent patient satisfaction can be shaped significantly through patient-centred-communication. The presented project represents part of the basic research in general medical care research and contributes to the transparent processing of theoretical assumptions. With the results described here, communication models with a focus on patient centredness can be evaluated with regard to their practical relevance and transferability.

**Funding:** This survey was part of a project that was funded by the German Federal Ministry of Education and Research (Bundesministerium fuer Bildung und Forschung, BMBF), grant number 01GY1605. The funders had no role in study design, data collection and analysis, decision to publish, or preparation of the manuscript.

**Competing interests:** The authors have declared that no competing interests exist.

## Introduction

In recent years, the demand for patient-centredness has gained importance in primary care as well in general medical care research [1,2]. If one focuses on the primary care consultation, it can be stated that it is largely determined by communication skills. One of the key challenges that General Practitioners (GPs) currently face is to improve the doctor-patient communication to the best intelligibility and reliability as well as the best quality possible [3]. If one now relies on the hypothesis that the satisfaction of a patient can be guaranteed by the communication, paternalistic ways of implanting communication are assumed to unlikely satisfy patients [4]. Particularly nowadays in general medicine, great value is given to patient-centred communication [5,6].

Furthermore, Scholl et al. identified in their study (*Integrative model of patient-centredness*) the doctor-patient-communication as one of their dimensions of patient-centredness. They concluded that communication functions as an enabler [2]. Regarding the patient's satisfaction with the communication, another study (*Commonwealth Fund)* found out that doctor-patient communication was assessed as deficient [7,8]. Another study dealing with patient-centred communication [9] tried to gain insights into the doctor-patient communication within clinics from the patient's perspective and dealt with the examination atmosphere, the comprehensibility of the doctor's statements, and the involvement in the decisions and the course of the conversation.

The relevance of examining patient satisfaction for the German primary care consultation arose from the *ICE study* [10] conducted by the research network *PRO PRICARE (Preventing Overdiagnosis in Primary Care)* of the Institute of General Practice of the University Hospital Erlangen. This cluster-randomised trial (CRT) focuses on the different medical communication methods and their implementation in the primary care consultation. One part of this study was to assess the patient's emotional reaction to the communication by a target-actual comparison of the GPs consultation in order to analyse the effect of patient-centred communication regarding chronic low back pain (LBP) [11]. Therefore, a new quantitative survey tool, the *Patient Satisfaction in Primary Care Consultation—Questionnaire* (*PiC*) was developed by the authors to analyse the following research question: *Does the implementation and application of patient-centred communication methods generate patient satisfaction in the primary care consultation*?.

Based on the *hypothesis* that patient satisfaction within the primary care consultation is anchored to communication with the GP, the following central assumption is made: The better patient-centred communication is implemented and used, the better the patients' satisfaction will be.

### Theoretical background

Regarding the German health care system, GPs occupy a special position as they are the first and most important point of contact for patients: The GPs act as a junction with a coordinating function in the medical care. This includes that they are available as the central point of contact for acute as well as continuous medical care of the patients. In accordance with the principles of participatory decision-making, the GPs assume responsibility for general medical care. No matter how much general medicine care changes, nothing will undermine the central role of the consultation between the GP and their patients. The importance gains even more weight in patient-centredness. Within the primary care consultation, the GP is the expert in the medical field with a deeper knowledge than the patient, visible in specialist terminology for instance. The consultation's aim is to find out and understand why an individual patient has sought help and advice [12]. For that, all communication needs to overcome improbabilities:

To communicate successfully, *receiving, understanding and accepting* a message are prerequisites in the interactional process [13]. In order to overcome these improbabilities in the GPs consultation, several medical communication concepts—so called *success media* [13]—were created and institutionalised [12].

## Concepts of medical communication

Based on the literary examination of various patient-centred communication methods, the ones presented below, emerged as the most widely used and best-known communication tools and therefore served as the basis for creating the *PiC-questionnaire* and its items. The concepts presented have developed within the medical discourse and partly exist side by side, but also build on each other because they are logically interwoven. As a first communication concept, *patient-centred communication (PCC)* itself places the consideration of the patient's perspective at the centre of the consultation and calls for a consideration of the patient from a biopsychosocial point of view. It entails understandable information about the diagnosis, treatment option and course of the disease with the focus on the aspect of language simplification. The most important element in the implementation of *PCC* is mutual listening and communicating between the GP and its patient. The GP should not only provide the patient with factual information, but also address the patient's individual experience and needs. In addition, the common understanding and the division of responsibility in the medical discussion should be clarified [14,15]. In the case of the second concept, *shared-decision-making (SDM)*, the expert competence lies not only with the doctor but also with the patient who plays an active role in the process of *making a decision*. Patients are given the opportunity to present their concepts of their illness and the possible treatment options in the consultation (*sharing ideas*). With this method, GP and patient should, if possible, come to a joint decision about treatment or therapy. This communication concept, which can also be found in the literature under the concept of participatory decision-making, was developed by Charles (1997) [16] and is an interaction process with the aim of sharing the active participation of patient and doctor with equal rights information to come to a jointly responsible agreement [16]. The objective of the third concept, the so-called *ideas, concerns, and expectations (ICE)* methodology according to Matthys [16], is to make patients aware of their personal ideas about their illness, their concerns about their symptoms, and their expectations of the treatment. Disclosing a patient's ideas, concerns, and expectations can not only provide better insight into the reasons why a patient is visiting the GP, but can also help make the correct diagnosis faster. In addition, it places demands on the process of shared-decision-making and can have a positive effect on the patient's cooperative behaviour in the context of their therapy and consequently optimise the patient's compliance. In summary, it can be stated that the *ICE* method is exposed as a further medium, which increases the likelihood of understanding but also the acceptance of communication in the GP consultation [17,18].

## Concept of patient satisfaction

The operationalisation of satisfaction of the *PiC* goes back to the *confirmation-disconfirmation-paradigm* (C/D paradigm) according to Yi (1990) [3,20], which deals with the recording of satisfaction resulting from a cognitive evaluation of a received achievement. Accordingly, the developed *PiC* is based on the following understanding of satisfaction: In order to be able to determine patient satisfaction as a measurable quantity in this study, it is of importance to first find out what is meant by *satisfaction* [19]. It can be assumed that every patient has certain individual expectations before a consultation, the fulfilment of which has a significant influence on their satisfaction. The construct of satisfaction is described as a cognitive assessment

of expectations and actual experience, as a spontaneous or emotional reaction [19]. The term *satisfaction*, therefore, involves the comparison between an expectation and the actual experience. From this, expectations can be seen as an indispensable premise for satisfaction by which the quality of a service is estimated. If an emotional evaluation evolves, implying a strong positive or strong negative difference, satisfaction or dissatisfaction arises. It is presumed that a patient compares the experienced performance with the expected performance of the consultation. If the expected performance corresponds to the experienced performance, the patient feels neutral towards the communication, meaning that the patient is neither satisfied nor dissatisfied, and the communication has neither had a positive nor a negative influence on the satisfaction of the patient [3,20].

## Materials and methods

The ICE study aims to discuss the influence of patient-centred communication on patient satisfaction. The ethical approval of the ICE study was granted by the Ethics Committee of the Faculty of Medicine of the Friedrich-Alexander University Erlangen-Nürnberg (296_17B, 18.10.2017). The ethics committee does not raise any objections to the conduct of the study and approved it. The form of consent of the ethics statement was obtained written. As part of the ICE study a universal primary care survey tool, the *PiC* was developed by the authors for the purposes of this study, which focuses on patient satisfaction in the context of patient-centred-communication. According to the ethics statement, the participants in the study do not need to obtain a written declaration of consent. Oral information and consent from the patient are sufficient.

The *PiC* is based on the *PREMS methodology (Patient Reported Experience Measures)*, which measures the perceived experience of the patient with the treatment process and was created on the basis of the target-actual-comparison according to Yi (1990) [3,20–22]. The aim of the *PiC* is not about evaluating the GPs, but rather to raise the issue to what extent communication really matters for shaping patient satisfaction in the primary care consultation and was therefore created for general medical advice purposes. Previous satisfaction surveys served as inspiration for the creation of the *PiC*, but did not meet the query of satisfaction based on a target-actual comparison, which underlines the necessity of a new developed questionnaire [3,20]. Items and modules of the *PiC* were operationalised and selected based on a literature research related to medical patient-centred communication methods. The *PiC* was created and validated in German language and programmed in a paper-based design with EvaSys Version 7.1 (S1 File). An unvalidated English version is also attached to this research paper as Supporting Information (S2 File).

### Structure and measures

The structure of the *PiC* is divided into four modules (A, B, C and D) which can be described as follows:

*Module A* examines the *current state of primary care consultation* with regard to the actual use of the medical patient-centred communication methods.

*Module B* explores *the expectation of the patient* about the applied communication methods in the GP consultation. In this way, the target-actual state of communication can be compared in a GP consultation.

The items of module A as well as of B are tabulated in a unipolar and ordinally scaled Likert scale: *(1) Fully agree, (2) Agree, (3) Disagree, (4) Disagree completely, (5) Do not know*. In *module A*, the patient was asked to *please choose which answer is best for him/her after today's conversation with his/her GP* (current state of consultation). *Module B* asks the patient to *please*

*choose what he/she expects from a medical consultation with his/her GP* (patient expectation regarding the consultation).

*Module C* concludes with a query of the overall satisfaction of the medical interview in the GP consultation: patients should be given the opportunity to make a clear statement about their satisfaction with the communication between them and the doctor in the GP consultation and were therefore asked: *How satisfied were you with today's general medical consultation*? To answer this, the scale is structured in a four-stage satisfaction/dissatisfaction scale: *(1) Very satisfied, (2) Satisfied, (3) Not satisfied, (4) Not satisfied at all, (5) Do not know*. Furthermore, *module C* also deals with the time aspect of a GP consultation.

Moreover, this is followed by the query of expectation where the patient is asked about h*ow much time he/she thinks should be planned for a doctor's consultation*, in order to assess his/her expectation (*target state*) on the time frame. A comparison of the target-actual state is determined by categorised response specifications with average values of the duration of general practitioner consultations. For that, the satisfaction with the time frame (*actual state*) was examined through the item: *How did you find the duration of today's doctor's consultation*? Participants were asked to indicate both their expectation as well as their satisfaction with the time frame on a five-stage two-dimensional scale.

The questionnaire ends in *module D* with additional demographic questions about the respondent's age, gender, and insurance status.

## Operationalisation of the items in module A and B

The operationalisation of the questions and statements in *module A and B* (points 2 and 3 in the questionnaire) arose from the theoretical components of the medical communication methods such as *PCC*, *SDM* and *ICE*, which contribute to overcoming the improbabilities of communication in the primary care consultation. Those items build on each other in an actual-target comparison and are arranged as follows in the questionnaire (S1 and S2 Files):

- C*ommunicative exchange* examines the substantive discussion with the GP during the consultation. This is primarily about whether a conversation with the GP has come about and a patient is interested in communicating (*PCC, items 2.1/3.1*).

- *Equipoise* calls for a balance between GP and patient in terms of joint decision-making. These items examine whether a patient has been taken seriously in a consultation and treated with equal rights, and whether a patient leaves out the opinion of the GP (*PCC, items 2.2/ 3.2*).

- *Educating* the patient means that the patient should be informed about all possible options as well as their advantages and disadvantages (*SDM, items 2.3, 2.4/3.3*).

- *Understandability* questions whether the treatment steps are explained comprehensibly (*SDM, items 2.5/3.4*).

- *Traceability* examines whether the patient can understand the treatment or not (*SDM, items 2.6/3.5*).

- *Discussability* highlights the GP's side, and whether he/she is ready to move away from his/ her point of view and allows a discussion with the patient (*SDM, items 2.7/3.6*).

- *Consent* specifically asks for satisfaction with the final decision on a treatment. Reciprocally, it is examined whom the patient trusts more in the decision-making process–the GP or his/ her own views (*SDM, items 2.8/3.7*).

- *Ideas*, *concerns*, *and expectations* analyse whether the patient's personal ideas were considered, concerns were addressed or whether a patient had the opportunity to present them to the doctor and finally deal with the expectations a patient puts on the handling of his/her symptoms (*ICE*, items 2.9–2.11, 3.8–3.10).

## Pretest

The newly developed questionnaire was pretested through cognitive interviews [22]. A statistical validation was not previously carried out. Accordingly, this study can be viewed as a pilot study. The created *PiC* was tested with 11 volunteers (7 patients and 4 GPs) of the medical care centre of the University Hospital Erlangen. The volunteers were asked directly at the medical care centre to test the paper-based questionnaire that had been developed. The tested volunteers were 64% female (n = 7) and were all over 18 years old. Thus, the comprehensibility of the questions and the associated instructions for completing were checked. Furthermore, an exchange took place about the meaning of the question and answer items and the comprehensibility of their formulations. As a result of this pretest, it was found that the structure of the questionnaire and the question formulations were comprehensible. Some adjustments were made concerning the wording and the graphic representation.

## Recruitment and data collection

The recruitment of the GPs in this CRT took place among German GPs belonging to four independent practice networks in Northern Bavaria/Germany. The building of the clusters involved two levels: GPs recruited to receive a training in patient-centred communication (*intervention group*) or no training (*control group*). To minimise contamination in the control group, randomisation to patient-centred communication training intervention took place at the practice level with units of randomisations being single-handed or group practices belonging to the research network *Forschungspraxen Franken* (a newly setup research network located in rural and urban areas of Franconia/Northern Bavaria comprising 119 GPs from 77 practices). This randomisation ensured that doctors being allocated to the control group will not be surrounded by colleagues having received the training intervention. Furthermore, all participants will be blinded towards the explicit purpose and design of the study. The communication training was offered to all participants: to GPs in the intervention group as a true intervention at the beginning of the CRT and to control GPs as a pretend intervention at the end of the trial. The clusters of the GPs were formed considering the number of participating GPs in a practice. This enabled a numerical balance between the intervention and control groups [11]. Accordingly, all GPs were randomly allocated to the respective study arm in intervention group with *5 GPs* and control group with *9 GPs*.

## Sample size calculation

Regarding the sample size calculations, it confirmed the need for sample size inflation due to the cluster design so that each GP could form a cluster of clinical decisions that contain similar treatment decisions that are not independent of one another. The sample sizes also considered the intra-cluster correlation coefficient, number of events, expected effect, and study performance. Assuming a referral rate of *30%* for acute LBP, as indicated in German routine data, an absolute change in referring patients of the order of *10%* was considered clinically relevant. Assuming an intra-cluster correlation coefficient (ICC) of *0.05*, a significance level of *0.05* and a potency of *0.8*, *24 GPs*, each with *40 patients* (*n = 1920*), are required in each study group to achieve a *30%* decrease in referrals in the Control group found to be *20%* or less in the

intervention group [11]. Alternative requirements for the sample size and the final sample were considered based on the actual referral rates and were adjusted accordingly. Therefore, the final sample consisted of *14 GPs in total* with *80 patients* each and a total n with *1120 patients* [11]. In total, the response rate lay at *79% (n = 880)*.

## Patient sampling

Concerning the patients sampling, the medical assistants of the GP practices were responsible for the transmission of patients and returning the *PiC-questionnaire* to the Institute of General Practice in Erlangen. The patient data collection took place by the medical assistant within the GP practice assuring patient anonymity of the collected data. Regarding the ethics statement a written informed consent from the patients was therefore not required. The data being transferred to the Institute of General Practice in Erlangen did not allow the detection of patient identity [11]. Every *third patient* (men and women) over the *age of 18* who presented themselves to their GP within the *three-month period* was included in the survey. Inclusion criteria were consultations involving patients consulting their doctor for uncomplicated acute LBP. After the consultation, the paper-based questionnaire was immediately filled out anonymously by the patient in the practice–none was handed home. The time frame of the data collection was from November 2018 to January 2019 and was collected by the practices within 4–8 weeks [10,11]. In general, the *PiC* can be used as a representative instrument for patients with general consultation needs in a GPs practice and does not refer to any specific sociodemographic sample.

## Intervention

Beyond that, the intervention group attended a training on patient-centred communication. This one-day training was held at the Institute of General Practice in Erlangen. The concept was designed in an open manner and was based on the methodology of group discussions. Thus, there was a high proportion of speeches by the participants and only stimuli from the training moderator were given. The training was aimed at a small number of participants in order to create the highest possible level of trust. The training consisted of an introductory round followed by a discussion in which the participating GPs could exchange information about their practiced communication. This was followed by a refresher of already known communication methods such as SDM and continued with learning about the ICE technique. The intervention group completed the training before receiving the questionnaire, the control group afterwards. In close association with recommendations of the national guidelines for acute LBP, the training provided clues on how to encourage patients to report their ICE and offers communication skills training through standardised patient scenarios [11].

## Analyses and results

All analyses of this study, as well as the described validation, were carried out with *IBM SPSS Statistics 24*. All relevant data and syntaxes are within the paper and its Supporting Information Files (S3–S5 Files).

## Validation

As the *PiC* is a new developed instrument, the following tests for validation were made and have proven the *PiC* to be a valid reliable instrument. The *factor analysis* shows the following results: *Kaiser-Meyer-Olkin criterion (KMO)* shows in our data set that it is *significant at 0.932*; i.e. the recommended values should be at a minimum of *0.5* [23]. That is followed by the

**Table 1. Factor analysis: KMO and Bartlett's Test.**

| KAISER-MEYER-OLKIN | BARTLETT'S TEST OF SPHERICITY | | |
|---|---|---|---|
| | $\chi^2$ | df | p |
| 0.932 | 5581.83 | 210 | 0.000 |

*Bartlett's sphericity test* that tests the *null hypothesis (H0)* of whether the correlation matrix is an identity matrix. In our data set there is a *high significance on the relationship between the variables at 0.000*. Results can be seen in Table 1.

The *factor analysis* in total shows a high load of individual items on the corresponding component: The query of the actual state (*actual items*) all load onto the same component 1 with a *37.067% variance*. The query of the target state (*target items*) all load onto the component 2 with a *12.373% variance*. Only one item (*target item*) loads incorrectly with a factor of *0.369* to component 1 instead of 2. Results can be seen in Table 2.

Furthermore, the internal consistency of the questionnaire is satisfying, with *Cronbach's alpha for positive affect 0.908*. Results are depicted in Table 3.

*Split-half reliability* shows that *Cronbach's Alpha* can be classified as *very good (target 0.905)* and *good (actual 0.842)* in both parts. The split-half reliability coefficient shows a *positive relationship with 0.526*. Results can also be seen in Table 4.

**Table 2. Factor analysis: Rotated component matrix[a].**

| | COMPONENT | |
|---|---|---|
| | 1 | 2 |
| Were your personal fears concerning your symptoms considered? | 0.775 | |
| Were your expectations concerning the handling of your symptoms considered? | 0.772 | |
| Were you able to discuss different treatment options with your GP? | 0.751 | |
| Were your ideas concerning your symptoms considered? | 0.736 | |
| Are you satisfied with the decision concerning your further treatment? | 0.726 | |
| Were you able to understand the treatment approach? | 0.719 | |
| Did you feel as though you were taken seriously by your GP? | 0.693 | |
| Did your GP explain the steps of the treatment in a comprehensive manner? | 0.680 | |
| Did your GP explain various treatment options? | 0.663 | |
| Did you feel as though you were able to talk to your GP? | 0.646 | |
| Did you want to know more about various treatment options? | 0.461 | 0.336 |
| I trust the decision making and the opinion of my GP fully. | 0.369 | 0.308 |
| I want to be able to understand the treatment procedure. | | 0.770 |
| The GP should explain all treatments comprehensively. | | 0.740 |
| I want to be able to present my expectations to my GP. | | 0.740 |
| The GP should explain all treatment options. | | 0.736 |
| The GP should consider my personal fears regarding my symptoms. | | 0.713 |
| I want to be able to discuss various treatment options with the GP. | | 0.669 |
| My own point of view should be worth at least as much as that of the GP. | | 0.583 |
| I should have the opportunity to talk to the GP. | | 0.545 |
| My GP should form their own opinion, regardless of my personal ideas concerning my symptoms. | | 0.496 |

*Rotation method*: *Varimax with Kaiser-Normalization[a]*.

a. Rotation converged in 3 iterations.

**Table 3. Reliability (Cronbach's α).**

| CRONBACH'S A | | N |
|---|---|---|
| | *based on standardised items* | |
| 0.908 | 0.913 | 21 |

In addition, external validation can be guaranteed due to the field limitations of the PiC (field access, sample etc.).

## Patient cohort characteristics

First of all, a descriptive analysis was made. As we had a total n with *880 patients* of the two clusters of the intervention and control group, the patient cohort characteristics can be found as follows in Table 5.

**Table 4. Item statistics and correlation (Cronbach's α, split-half).**

| | Scale mean *if item deleted* | Scale variance *if item deleted* | Correlation *r* | Cronbach's *α* |
|---|---|---|---|---|
| **A. Actual Status** | | | | |
| 1. Did you feel as though you were able to talk to your GP? | 73.97 | 34.33 | 0.474 | 0.906 |
| 2. Did you feel as though you were taken seriously by your GP? | 73.98 | 33.99 | 0.520 | 0.905 |
| 3. Did your GP explain various treatment options? | 74.16 | 32.59 | 0.579 | 0.903 |
| 4. Did you want to know more about various treatment options? | 74.27 | 32.62 | 0.518 | 0.905 |
| 5. Did your GP explain the steps of the treatment in a comprehensive manner? | 74.08 | 33.11 | 0.585 | 0.903 |
| 6. Were you able to understand the treatment approach? | 74.11 | 32.69 | 0.654 | 0.901 |
| 7. Were you able to discuss different treatment options with your GP? | 74.21 | 31.81 | 0.675 | 0.900 |
| 8. Are you satisfied with the decision concerning your further treatment? | 74.08 | 33.06 | 0.594 | 0.903 |
| 9. Were your ideas concerning your symptoms considered? | 74.17 | 32.59 | 0.617 | 0.902 |
| 10. Were your personal fears concerning your symptoms considered? | 74.14 | 32.48 | 0.647 | 0.901 |
| 11. Were your expectations concerning the handling of your symptoms considered? | 74.16 | 32.53 | 0.627 | 0.902 |
| *Summary A* | | | | *0.905* |
| **B. Target Status** | | | | |
| 1. I should have the opportunity to talk to the GP. | 73.98 | 34.30 | 0.432 | 0.906 |
| 2. My own point of view should be worth at least as much as that of the GP. | 74.45 | 32.58 | 0.431 | 0.908 |
| 3. The GP should explain all treatment options. | 74.08 | 33.14 | 0.571 | 0.903 |
| 4. The GP should explain all treatments comprehensively. | 74.01 | 33.67 | 0.549 | 0.904 |
| 5. I want to be able to understand the treatment procedure. | 74.09 | 33.28 | 0.504 | 0.905 |
| 6. I want to be able to discuss various treatment options with the GP. | 74.18 | 32.95 | 0.530 | 0.904 |
| 7. I trust the decision making and the opinion of my GP fully. | 74.37 | 32.42 | 0.435 | 0.908 |
| 8. My GP should form their own opinion, regardless of my personal ideas concerning my symptoms. | 74.14 | 33.52 | 0.442 | 0.906 |
| 9. The GP should consider my personal fears regarding my symptoms. | 74.09 | 33.08 | 0.576 | 0.903 |
| 10. I want to be able to present my expectations to my GP. | 74.23 | 32.39 | 0.572 | 0.903 |
| *Summary B* | | | | *0.843* |
| | **Guttman Split-Half Coefficient** | | **Spearman-Brown Coefficient** | **Correlation between forms** |
| **Overall** | 0.686 | | 0.689 | 0.526 |

**Table 5. Patient cohort characteristics.**

| | | N | | | % | | |
|---|---|---|---|---|---|---|---|
| | | **Control** | **Intervention** | **Total** | **Control** | **Intervention** | **Total** |
| | | 309 | 571 | 880 | 35,1% | 64,9% | 100% |
| Gender | Female | 174 | 356 | 519 | 57% | 62% | 59% |
| | Male | 135 | 215 | 361 | 43% | 38% | 41% |
| Mean age | | 59 | 58 | 59 | | | |
| Statutory health insurance | | 255 | 497 | 810 | 83% | 87% | 92% |

However, the focus lies in the second step which was to create a satisfaction index that filters satisfaction out of the target-actual comparison of *modules A and B* of the *PiC* regarding the intervention and control group. Finally, possible influencing on patient satisfaction were examined using a linear regression.

## Target-actual comparison

In order to reveal a connection between patient-centred communication and patient satisfaction with the medical consultation, the present sample was divided into an intervention group and a control group. The interesting feature of satisfaction is measured by an *additive index (satisfaction_is_ind, satisfaction_should_ind)*, which was artificially defined for both groups in the *value range -10 to +10*, not at all satisfied to maximally satisfied (S4 File). In order to be able to show the difference between the target-actual comparison of *modules A (actual state)* and *B (target state/patients' expectation)* of the PiC and thus the satisfaction in this comparison, a dummy variable (*overall_difference_satisfaction*) was built that answered the difference between the target-actual comparison (*differently answered Items of modules A and B*). For this purpose, the total number of points for the items in both modules was weighted with the individual maximum possible number of points (S5 File). The aim of this variable is to find out what is in the 'neutral' range (*0: no difference between the answers to module A and B*) or

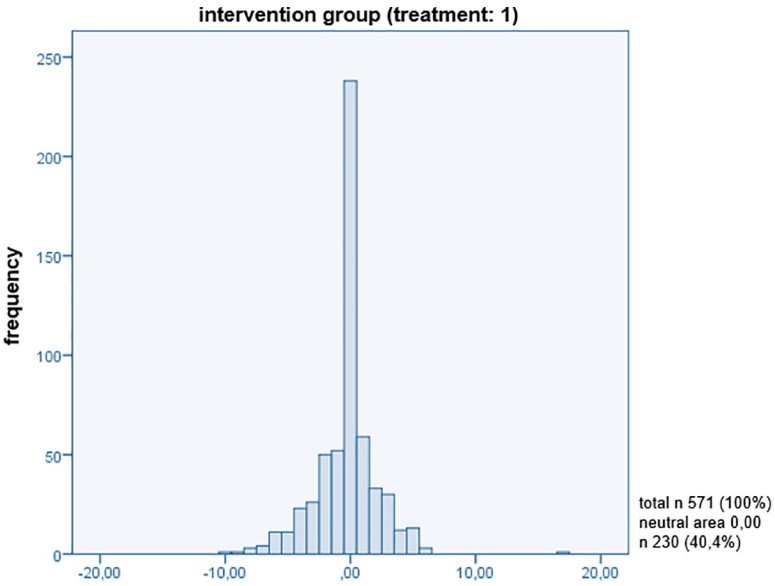

**Fig 1. Intervention group (treatment: 1).**

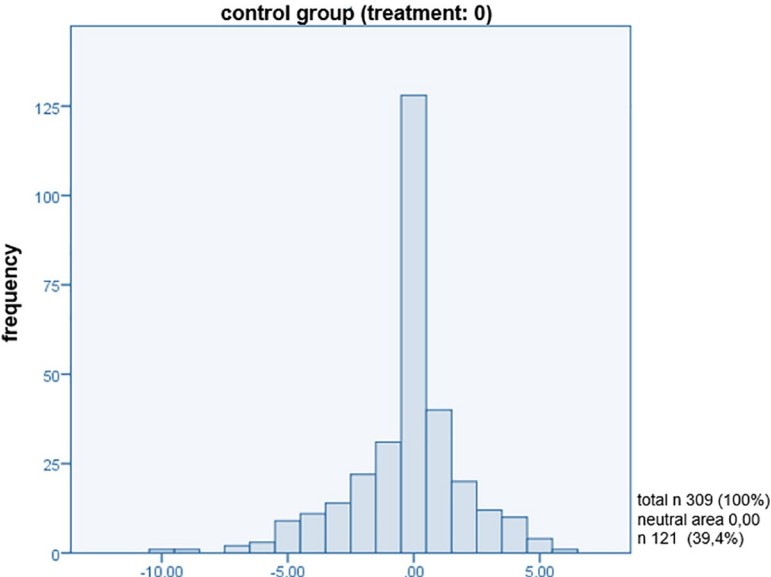

**Fig 2. Control group (treatment: 0).**

outside this range *(-10 > 0 < +10)*. This index was applied to the cluster's intervention and control group individually and to both clusters together. Values that are within the negative range of *-10 to <0* indicate that the patient's expectations did not meet the expected communication. If values are within the range of *>0 to +10*, the patient's expectations *had been exceeded*. If the value reaches *0*, the patients are in the neutral range and were neither extremely satisfied (*expectation was not exceeded*) nor unsatisfied (*expectation was not met*) with the communication in the consultation. This means that the patient had no great expectations of communication during the consultation and did not attribute as high of an importance to it as was expected in our hypothesis. It was established that *40.4%* of the respondents in the intervention group (*treatment*: *1*) were in the neutral area (Fig 1). In comparison to that, *39.4%* of the control group (*treatment*: *0*) were also in the neutral area (Fig 2). The analysis is based on these percentages of the different answers given (*target-actual comparison*) in modules A and B with regard to patient-centred communication, and it is precisely for this reason that this new approach to querying patient satisfaction is presentable. On the one hand, this could make clear that the training had no influence on the application of the communication methods of the GPs. On the other hand, contrary to the assumption that a smaller distribution of patients would be in the neutral area, this could mean that communication is not assigned the highest priority on the patient's side. The assignment of values in the negative and positive range shows an even distribution within a group and no differences between the clustered groups (Fig 3). There were no differences found between the intervention and control groups in the present sample. Histograms of the target-actual comparison are shown below in Figs 1–3.

## Influencing factors

As there were no differences found between the intervention and control groups regarding the target-actual comparison in the present sample, a linear regression model for both groups without subdivision of the clusters was carried out to find factors that could be influencing the patient. In relation to the regression model, various models were calculated and the one with

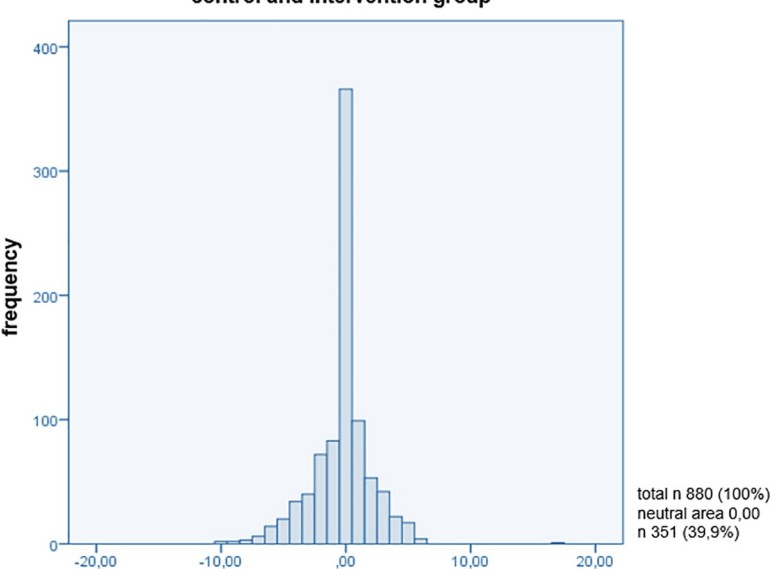

**Fig 3. Control and intervention group.**

the best model quality ($R^2$) and the influencing variables of interest in this study were used. Accordingly, post-hoc corrections of various models were carried out in this regard. We found out that socio-demographic characteristics could have an impact on the patient satisfaction (*regression coefficient and dependent variable*: *dummy variable overall_difference_satisfaction*). Due to the low model quality ($R^2$: *3.5%*) of the linear regression, we could only deliver hypotheses which indicate that patient-centred communication is not the only factor influencing patient satisfaction. In the linear regression, it was possible to determine that the older a patient (*age*) is (*p = 0.017*), the more satisfied with the implementation of patient-centred communication they are and that their expectation was exceeded (>0: *positive range of the target-actual comparison*). Furthermore, men (*p = 0.007*) were more dissatisfied than women, meaning that their expectation was not met (<0: *negative range of the target-actual comparison*). Regarding the time aspect (*time_consultation*), it can be shown that patients who found the duration of the GPs consultation exactly right proved to be more satisfied (*p = 0.000*). They tend to feel that their expectations were exceeded regarding the conversation (>0: *positive range of the target-actual comparison*) and are therefore more satisfied. The insurance status (private) had *no significant influence (p = 0.342)* on the patient's satisfaction. Results can be found in Table 6.

## Discussion and conclusion

To conclude, it can be stated that patient satisfaction cannot solely be attributed to patient-centred communication. Nevertheless, the satisfaction effect here can only be derived from our research results. The aim was to scientifically work through the dogma of patient satisfaction in the GP consultation. The hypothesis that patient satisfaction can be only guaranteed by the application of patient-centred communication methods cannot be confirmed in this study. On the one hand, we could assume that the training on patient-centred communication had no influence on the application of the communication of the GPs. On the other hand, contrary to the assumption that a smaller distribution of patients would be in the neutral area, it could mean that communication is not assigned the highest priority from the patient's side.

**Table 6. Regression model.**

| A. Model Summary[b] | | | | | |
|---|---|---|---|---|---|
| | **Correlation Coefficient *R*** | ***R*²** | **Adjusted *R*²** | | |
| | 0.186[a] | 0.035 | 0.030 | | |
| **B. Predictor coefficients[b]** | | | | | |
| | Unstandardised coefficients | | Standardised coefficient | | |
| | **Regression Coefficient *B*** | **Standardised error** | | | |
| (Constant) | 9.912 | 0.998 | | $t = 9.934$ | $p = 0.000$ |
| Age | 0.030 | 0.013 | $\beta = 0.083$ | $t = 2.384$ | $p = 0.017$ |
| Man | -1.197 | 0.447 | $\beta = -0.094$ | $t = -2.681$ | $p = 0.007$ |
| Private insurance | 0.736 | 0.775 | $\beta = 0.033$ | $t = 0.950$ | $p = 0.342$ |
| Time aspect | 3.084 | 0.816 | $\beta = 0.133$ | $t = 3.781$ | $p = 0.000$ |

a. Predictors: (Constant), time consultation, private, age, man.

b. Dependent variable: Overall_difference_satisfaction.

Furthermore, another confounder could be that other communication trainings had already been applied to the tested groups and the GPs therefore had not taken full advantage of the current training or would rather rely on their abilities to communicate intuitively.

Limitations of this study can be seen in the examination of patient satisfaction in the consultation: patient satisfaction cannot be related to patient-centred communication alone. Furthermore, there are limitations in the operationalisation of the items, which were obtained purely from the theoretical patient-centred communication methods.

However, strengths can be seen in the development of this new survey tool, which put the patient-centred communication methods between GPs and patients in the foreground. In general, this study breaks down patient-centred communication, as it is taught, into details and queries them in a compact format. Moreover, this is one of the few studies which tries to evaluate the application of patient-centred communication by the GPs from the patient perspectives.

Moreover, the results of this study show that the patient's satisfaction does not depend solely on the patient-centred communication with the GP. The aim of the study was to reveal the quality of primary care consultation by putting the patient-centred communication in the focus. Furthermore, the goal was to implement new patient-centred communication models like the ICE technique through the training on patient-centred communication (intervention), and to further evaluate existing ones from a patient perspective with the *PiC*. It can be stated that the call for the implementation of patient-centred communication in practice as well as in medical education and training is very strong. However, on the basis of this survey, other important factors can be assumed that influence the satisfaction of the patient. Accordingly, this research paper puts the need for patient-centred communication up for discussion. We conclude that the aim is not to increase or improve the quality of communication between GP and its patient, but rather to adapt the structures and the framework of communication in the GP's practice. In summary, we can say that the situation and the communication in the GP consultation can be viewed as a highly institutionalised context in which different fields and influences meet. It can be assumed that patient-centred communication could improve patient's satisfaction, but is not decisive. Therefore, other factors must also be considered such as a long and stable doctor-patient relationship, but also the connection to the GP's practice, the opening times and the treatment spectrum–to name just a few.

## Supporting information

**S1 File. German validated version of the PiC.**
(PDF)

**S2 File. English unvalidated version of the PiC.**
(PDF)

**S3 File. Data.**
(SAV)

**S4 File. Syntax.**
(SPS)

**S5 File. Syntax.**
(SPS)

## Acknowledgments

The present work was performed in fulfilment of the requirements for obtaining the degree "Dr. rer. biol. hum." for Stefanie Stark.

## Author Contributions

**Conceptualization:** Larissa Burggraf.

**Formal analysis:** Lukas Worm.

**Methodology:** Stefanie Stark, Marco Roos, Larissa Burggraf.

**Supervision:** Marco Roos, Larissa Burggraf.

**Validation:** Stefanie Stark.

**Writing – original draft:** Stefanie Stark.

**Writing – review & editing:** Marie Kluge, Marco Roos, Larissa Burggraf.

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
