## [Decision Letter · Decision Letter 0]

13 Apr 2021

PONE-D-20-37750

The Patient Satisfaction in Primary Care Consultation - Questionnaire (PiC): an Instrument to assess the impact of Patient-centred communication on Patient satisfaction

PLOS ONE

Dear Dr. Stark,

Thank you for submitting your manuscript to PLOS ONE. After careful consideration, we feel that it has merit but does not fully meet PLOS ONE’s publication criteria as it currently stands. Therefore, we invite you to submit a revised version of the manuscript that addresses the points raised during the review process.

We look forward to receiving your revised manuscript.

Kind regards,

Paola Gremigni, Ph.D.

Academic Editor

PLOS ONE

Journal Requirements:

2. In the ethics statement in the Methods section and online submission information, please specify the type of informed consent that was obtained from the participants (for instance, written or verbal, and if verbal, how it was documented and witnessed)."

3. Please include additional information regarding the survey or questionnaire used in the study and ensure that you have provided sufficient details that others could replicate the analyses. For instance, if you developed a questionnaire as part of this study and it is not under a copyright more restrictive than CC-BY, please include a copy, in both the original language and English, as Supporting Information, or include a citation if it has been published previously.

4. In your Methods section, please provide additional information about the participant recruitment method and the demographic details of your participants. Please ensure you have provided sufficient details to replicate the analyses such as: a) the recruitment date range (month and year), b) a description of any inclusion/exclusion criteria that were applied to participant recruitment, c) a statement as to whether your sample can be considered representative of a larger population, d) a description of how participants were recruited, and e) descriptions of where participants were recruited and where the research took place.

5. Please provide further details on sample size and power calculations.

6. In the Methods section, please clarify whether intra-cluster correlation was considered during  estimation of the effective sample size - given that different localities were sampled. Further, in the Statistical analysis section, please elaborate how you accounted for clustering by locality in your statistical models.

7.In statistical methods, please refer to any post-hoc corrections to correct for multiple comparisons during your statistical analyses. If these were not performed please justify the reasons. Please refer to our statistical reporting guidelines for assistance (https://journals.plos.org/plosone/s/submission-guidelines.#loc-statistical-reporting).

8. Thank you for stating the following in the Funding Section of your manuscript:

This survey was part of a project that was funded by the German Federal Ministry of

Education and Research (Bundesministerium für Bildung und Forschung, BMBF), grant number 01GY1605.

No. The funders had no role in study design, data collection and analysis, decision to publish, or preparation of the manuscript.

Reviewers' comments:

Reviewer's Responses to Questions

**Comments to the Author**

1. Is the manuscript technically sound, and do the data support the conclusions?

Reviewer #1: No

Reviewer #2: Yes

2. Has the statistical analysis been performed appropriately and rigorously? 

Reviewer #1: No

Reviewer #2: Yes

3. Have the authors made all data underlying the findings in their manuscript fully available?

Reviewer #1: No

Reviewer #2: Yes

4. Is the manuscript presented in an intelligible fashion and written in standard English?

Reviewer #1: Yes

Reviewer #2: Yes

5. Review Comments to the Author

Reviewer #1: The present study aimed to discuss the influence of communication on patient satisfaction and developed a universal primary care survey tool that focuses on patient satisfaction in the context of patient-centred-communication. Using a sample of consisting of 14 GPs with 80 patients each (n=1120), the Authors showed that correlation between patient-centred-communication and patient satisfaction was non-signifcant. Hence, their results raise the issue to what extent communication really matters for shaping patient satisfaction.

The focus on the link “patient-centred communication” -> “patient satisfaction” makes the paper very interesting, and it may be the strength of the article. However, I also have some concerns that I will present in what follows. They are indexed according to the structure of the paper.

1.I suggest to specify what GP means

2.In regards to this sentence “The PiC is also based on the PREMS methodology….”: The Pic should be previously introduced (the use of “also” suggests that it was, but this is not the case). Is this an instrument developed by authors? Does it derive from other instruments? Were content and face validity previously attested? If yes, in which way? How were items and modules chosen? I think that a brief description of these information should be provided before the “materials” section.

3.Section 3.5: The content of this section should be placed in “Results” section. Yet, I suggest to include in the paper tables regarding factor loadings, specific items, and name of the factors. Finally, I suggest the use of factorial analysis instead of principal component analysis.

4.More importantly, I suggest to provide evidence about the external validity of the instrument (even if based on previous findings).

5.In regards to this sentence at p. 8 “the analyses of modules A (current state) and B (patient expectation) were formed into a satisfaction index with a range of -10 to +10. Values that are within the negative range of -10 to <0 indicate that the patient's expectations did not meet the expected communication”: I suggest to specify how this index was developed/computed.

6.Section 4.2 “Influencing factors”: I suggest to report a table for regression results. Yet, it is not clear (at least for me) which is the regression coefficient for the relationship between “patient-centred communication” and “satisfaction”. Maybe the relationship between patient-centred communication” and “satisfaction” was only investigated by means of the results reported at the bottom of p.8. If yes, a more accurate analysis should be conducted, given that the analyses are solely based on a match between percentages (without any reference to effect sizes and significant tests).

7.In regards to this sentence in Discussion section “The aim of the study was to improve the quality of primary care consultation by improving the doctor-patient communication, to implement new patient-centred communication models (e.g. ICE) in practice, and to further develop existing ones from a patient perspective.”. I suggest to be more specific in explaining how this study addressed this aim. For example, the authors may outline which findings may be used to improve the quality of primary care consultation.

I hope that my suggestions (or some of my suggestions) may help Authors in improving their paper.

Reviewer #2: Dear author,

This is a good effort to improve patient-doctor communication. I agree that patient satisfaction is not only restricted to patient-centered communication but may be influenced by other factors.

However, it is important to improve the way a doctor communicates with his/her patients. Maybe the author can add a bit on the background of patient-centered communication training in the GP training program or settings in the author's country.

I also suggest to change the sentences in line 194 and 195 to past tense; ie "should be" and "should take place" .

Lastly, the supporting information was in German Language. I would appreciate if the author could also include the questionnaire used in English language.

6. PLOS authors have the option to publish the peer review history of their article (what does this mean?). If published, this will include your full peer review and any attached files.

Reviewer #1: No

Reviewer #2: No

---

## [Author Response · Author response to Decision Letter 0]

28 Jun 2021

Response to Reviewers

Journal Requirements:

Dear Dr. Gremigni thank you for your comments. Please find our answers below your comments:

- Style requirements as well as the file naming was adjusted. 

2. In the ethics statement in the Methods section and online submission information, please specify the type of informed consent that was obtained from the participants (for instance, written or verbal, and if verbal, how it was documented and witnessed)."

- The type of informed consent was added in the ‘Materials and methods’ section (p. 7-12): “Informed consent will […] not be required. The data being transferred to the Institute of General Practice in Erlangen will not allow detecting patient identity” (p. 11).

3. Please include additional information regarding the survey or questionnaire used in the study and ensure that you have provided sufficient details that others could replicate the analyses. For instance, if you developed a questionnaire as part of this study and it is not under a copyright more restrictive than CC-BY, please include a copy, in both the original language and English, as Supporting Information, or include a citation if it has been published previously.

- Additional information regarding the survey and the development of the PiC-questionnaire were added in the ‘Materials and methods’ section (p. 7-12): Especially see the ‘Structure and measures’ section (p.7-8) as well as the ‘Operationalisation of the items in module A and B’ section (p. 8-9). Furthermore, we included the developed questionnaire in the original language (German) as a validated sample version to this manuscript (S1 File) as well as the translated unvalidated version in English (S2 File).

4. In your Methods section, please provide additional information about the participant recruitment method and the demographic details of your participants. Please ensure you have provided sufficient details to replicate the analyses such as: a) the recruitment date range (month and year), b) a description of any inclusion/exclusion criteria that were applied to participant recruitment, c) a statement as to whether your sample can be considered representative of a larger population, d) a description of how participants were recruited, and e) descriptions of where participants were recruited and where the research took place.

- We added additional information about the participant recruitment method and the demographic details of our participants of the survey as well as the details to replicate our analyses in the ‘Materials and method’ section (p. 7-12): Especially the ‘Recruitment and data collection’ section (p.10), the ‘Patient sampling’ section (p. 11) and the ‘Intervention’ section (p.12) were newly added to the manuscript and the processes were described extensively in order to fulfil the listed points a) to d).

5. Please provide further details on sample size and power calculations.

- Further details on sample size and power calculations can be found in the ‘Materials and method’ section (p. 7-12): Please see the new added ‘Sample size calculation’ section (p. 11). 

6. In the Methods section, please clarify whether intra-cluster correlation was considered during estimation of the effective sample size - given that different localities were sampled. Further, in the Statistical analysis section, please elaborate how you accounted for clustering by locality in your statistical models. 

- Further details on the intra-cluster correlation were added in the ‘Materials and methods’ section (p. 7-12): Especially in the ‘Recruitment and data collection’ section (p. 10) and in the ‘Sample size calculation’ section (p. 11): “Assuming an intra-cluster correlation coefficient (ICC) of 0.05, a significance level of 0.05 and a potency of 0.8, 24 GPs, each with 40 patients (n=1920), are required in each study group to achieve a 30% decrease in referrals in the Control group found to be 20% or less in the intervention group” (p. 11).

Furthermore, in the ‘Analyses and results’ section (p. 12-18), especially in the ‘Influencing factors’ section (p. 17-18) we added how we accounted for clustering: “As there were no differences found between the intervention and control groups regarding the target-actual comparison in the present sample, a linear regression model for both groups without subdivision of the clusters was carried out to find factors that could be influencing the patient (p. 17). Please also see the ‘Target-actual comparison’ section (p.16-17) for further details.

7.In statistical methods, please refer to any post-hoc corrections to correct for multiple comparisons during your statistical analyses. If these were not performed please justify the reasons. 

- Please see the ‘Influencing factors’ section (p. 17-18): “In relation to the regression model, various models were calculated and the one with the best model quality (R²) and the influencing variables of interest in this study were used. Accordingly, post-hoc corrections of various models were carried out in this regard” (p.17).

8. Thank you for stating the following in the Funding Section of your manuscript:

This survey was part of a project that was funded by the German Federal Ministry of Education and Research (Bundesministerium fuer Bildung und Forschung, BMBF), grant number 01GY1605. We note that you have provided funding information that is not currently declared in your Funding Statement. However, funding information should not appear in the Acknowledgments section or other areas of your manuscript. We will only publish funding information present in the Funding Statement section of the online submission form. Please remove any funding-related text from the manuscript and let us know how you would like to update your Funding Statement. Currently, your Funding Statement reads as follows: No. The funders had no role in study design, data collection and analysis, decision to publish, or preparation of the manuscript. Please include your amended statements within your cover letter; we will change the online submission form on your behalf.

- Nothing has to be changed. We deleted the funding information from the Manuscript.

Reviewer Comments:

Reviewer #1

Dear reviewer #1 thank you for your comments. Please find our answers below your comments:

1. I suggest to specify what GP means.

- The abbreviation GP stands for general practitioner and is now added to the manuscript as well explained in the context of the German health care system and in the context of this study. Please also see the ‘Introduction’ section (p. 3, line 50ff) as well as the ‘Theoretical background’ section (p.4, line 81ff).

2. In regards to this sentence “The PiC is also based on the PREMS methodology….”: The Pic should be previously introduced (the use of “also” suggests that it was, but this is not the case). 

Is this an instrument developed by authors? Does it derive from other instruments? Were content and face validity previously attested? If yes, in which way? How were items and modules chosen? 

I think that a brief description of these information should be provided before the “materials” section. briefly summarised in front of "Materials".

- The listed sentence has been reformulated and the questions that have been raised were all answered in a briefly summary on p.7. In addition, further development steps of the questionnaire were added and described in detail in order to clarify the development process of the created PiC-questionnaire and to make clear that it is a new survey instrument developed by the authors. For further details please see the ‘Materials and methods’ section (p. 7-9), as well as the ‘Concepts of medical communication’ section (p. 5) and the ‘Concept of patient satisfaction’ section (p. 6) which function as the theoretical fundament of the development of the PiC.

3. Section 3.5: The content of this section should be placed in “Results” section. Yet, I suggest to include in the paper tables regarding factor loadings, specific items, and name of the factors. Finally, I suggest the use of factorial analysis instead of principal component analysis. 

- We replaced the ‘Validation’ section (section 3.5) to the ‘Analyses and results’ section (p. 12-18), you can now find it on p.12. Moreover, we included all necessary tables of the validation into the manuscript in order to be able to comprehend our validation steps and results, especially the steps of the factor analyses. It was of interest to us to see which item loads onto which component (actual-target comparison / modules A and B / component 1 and 2). Therefore, as a last step of the factor analyses, it was of consistent to us to interpret the component analyses. For further details please see p. 12-15.

4. More importantly, I suggest to provide evidence about the external validity of the instrument (even if based on previous findings). 

- External validation can be guaranteed due to the field limitations of the PiC-questionnaire like e.g. field access, sample etc. For further added details please see ‘Materials and method’ section (p. 7-12): Especially the ‘Recruitment and data collection’ section (p.10) and the ‘Patient sampling’ section (p. 11) contain further details that justify the external validity. Please also see p. 12, line 284ff: “[…] the PiC can be used as a representative instrument for patients with general consultation needs in a GPs practice and does not refer to any specific sociodemographic sample.”

5. In regards to this sentence at p. 8 “the analyses of modules A (current state) and B (patient expectation) were formed into a satisfaction index with a range of -10 to +10. Values that are within the negative range of -10 to <0 indicate that the patient's expectations did not meet the expected communication”: I suggest to specify how this index was developed/computed.

- The index formation and calculation have been supplemented and described in detail under the ‘Target-actual comparison’ section (p. 16-17). In summary, the index was formed as follows and satisfaction was determined as follows: The interesting feature of satisfaction is measured by an additive index, which was artificially defined for both groups in the value range -10 to 10, not at all satisfied to maximally satisfied. In order to be able to show the difference between the target-actual comparison of modules A (actual state) and B (target state/patients’ expectation) of the PiC and thus the satisfaction in this comparison, a dummy variable (overall_difference_satisfaction) was built that answered the difference between the target-actual comparison (differently answered Items of modules A and B). For this purpose, the total number of points for the items in both modules was weighted with the individual maximum possible number of points. The aim of this variable is to find out what is in the "neutral" range (0: no difference between the answers to module A and B) or outside this range (-10 > 0 < +10). This index was applied to the cluster’s intervention and control group individually and to both clusters together. 

For further details please also see the new added Histograms (p. 17, Fig 1- Fig 3): “It was established that 40.4% of the respondents in the intervention group (treatment: 1) were in the neutral area (Fig 1). In comparison to that, 39.4% of the control group (treatment: 0) were also in the neutral area (Fig 2). The analysis is based on this percentages of the different answers given (target-actual comparison) in modules A and B with regard to patient-centered communication, and it is precisely for this reason that this new approach to querying patient satisfaction is presentable. […] The assignment of values in the negative and positive range shows an even distribution within a group and no differences between the clustered groups (Fig 3). There were no differences found between the intervention and control groups in the present sample” (p.16-17).

6. Section 4.2 “Influencing factors”: I suggest to report a table for regression results. Yet, it is not clear (at least for me) which is the regression coefficient for the relationship between “patient-centred communication” and “satisfaction”. Maybe the relationship between patient-centred communication” and “satisfaction” was only investigated by means of the results reported at the bottom of p.8. If yes, a more accurate analysis should be conducted, given that the analyses are solely based on a match between percentages (without any reference to effect sizes and significant tests).

- A table of the regression model was added in the ‘Influencing factors’ section (p. 17-18). Moreover, the linear regression and its coefficients are now described in detail. Please also see p. 16 that explains: “In order to be able to show the difference between the target-actual comparison of modules A (actual state) and B (target state/patients’ expectation) of the PiC and thus the satisfaction in this comparison, a dummy variable (overall_difference_satisfaction) was built that answered the difference between the target-actual comparison (differently answered Items of modules A and B)”. The dummy variable function as the dependent variable in this regression model.

7. In regards to this sentence in Discussion section “The aim of the study was to improve the quality of primary care consultation by improving the doctor-patient communication, to implement new patient-centred communication models (e.g. ICE) in practice, and to further develop existing ones from a patient perspective”. I suggest to be more specific in explaining how this study addressed this aim. For example, the authors may outline which findings may be used to improve the quality of primary care consultation.

- Please see the ‘Discussion and conclusion’ section (p. 18-20) which now includes following: “Moreover, the results of this study show that the patient's satisfaction does not depend solely on the patient-centred communication with the GP. The aim of this study was to reveal the quality of primary care consultation by putting the patient-centered communication in the focus. Furthermore, to implement new patient-centred communication models like the ICE technique through the training on patient-centred communication (intervention), and to further evaluate existing ones from a patient perspective with the PiC. It can be stated that the call for the implementation of patient-centered communication in practice as well as in medical education and training is very strong. However, on the basis of this survey, other important factors can be assumed that influence the satisfaction of the patient. Accordingly, this research paper puts the need of patient-centered communication up for discussion. We conclude that the aim is not to increase or improve the quality of communication between GP and its patient, but to adapt the structures and the framework of communication in the GP’s practice. In summary, we can say that the situation and the communication in the GP consultation can be viewed as a highly institutionalised context in which different fields and influences meet. It can be assumed that patient-centred communication could improve patient’s satisfaction but is not decisive. Therefore, other factors must also be considered such as a long and good doctor-patient relationship, but also the connection of the GP’s practice, the opening times and the treatment spectrum – to name just a few” (p. 19-20).

Reviewer #2:

Dear reviewer #2 thank you for your comments. Please find our answers below your comments:

Maybe the author can add a bit on the background of patient-centered communication training in the GP training program or settings in the author's country. I also suggest to change the sentences in line 194 and 195 to past tense; ie "should be" and "should take place". Lastly, the supporting information was in German Language. I would appreciate if the author could also include the questionnaire used in English language.

-We added more details on the patient-centered-communication training in the new added section ‘Intervention’ section (p.13-14): “Beyond the intervention group attended a training on patient-centred communication. This one-day training was held at the Institute of General Practice in Erlangen. The concept was designed in an open manner and was based on the methodology of group discussions. Thus, there was a high proportion of speeches by the participants and only stimuli from the training moderator was given. The training was aimed at a small number of participants in order to create the highest possible level of trust. The training consisted of an introductory round followed by a discussion in which the participating GPs could exchange information about their practiced communication. This was followed by a refresher of already known communication methods such as SDM and continued with learning about the ICE technique. The intervention group did the training before receiving the questionnaire, the control group afterwards. In close association with recommendations of the national guidelines for acute LBP, the training provided clues on how to encourage patients to report their ICE and offers communication skills training through standardised patient scenarios.”

We also added some details on the settings regarding the country and put the GP into the context of the German health care system. Please see p. 3, line 65ff and the ‘Theoretical background’ section (p.4, line 81ff).

Furthermore, the sentence in line 194-195 was deleted due to the adjustment of the entire paragraph. For further details please see the ‘Pretest’ section on p. 9-10.

The supporting information now includes the developed questionnaire in the original language (German) as a validated sample version (S1 File) as well as the translated unvalidated version in English (S2 File).

---

## [Editor Report · Decision Letter 1]

1 Jul 2021

The patient satisfaction in primary care consultation - questionnaire (PiC): an instrument to assess the impact of patient-centred communication on patient satisfaction

PONE-D-20-37750R1

Dear Dr. Stark,

We’re pleased to inform you that your manuscript has been judged scientifically suitable for publication and will be formally accepted for publication once it meets all outstanding technical requirements.

Kind regards,

Paola Gremigni, Ph.D.

Academic Editor

PLOS ONE
---

## [Editor Report · Acceptance letter]

5 Jul 2021

PONE-D-20-37750R1 

The patient satisfaction in primary care consultation - questionnaire (PiC): an instrument to assess the impact of patient-centred communication on patient satisfaction 

Dear Dr. Stark:

I'm pleased to inform you that your manuscript has been deemed suitable for publication in PLOS ONE. Congratulations! Your manuscript is now with our production department. 

Kind regards, 

on behalf of

Prof. Paola Gremigni 

Academic Editor

PLOS ONE